# Conventional and Resin-Modified Glass Ionomer Cement Surface Characteristics after Acidic Challenges

**DOI:** 10.3390/biomedicines10071755

**Published:** 2022-07-21

**Authors:** Irina Nica, Simona Stoleriu, Alexandru Iovan, Ionuț Tărăboanță, Galina Pancu, Nicoleta Tofan, Răzvan Brânzan, Sorin Andrian

**Affiliations:** Faculty of Dental Medicine, Grigore T. Popa University of Medicine and Pharmacy, 700115 Iasi, Romania; nica.irina@umfiasi.ro (I.N.); ionut-taraboanta@umfiasi.ro (I.T.); galina.pancu@umfiasi.ro (G.P.); nicoleta.tofan@umfiasi.ro (N.T.); razvan.branzan@umfiasi.ro (R.B.); sorin.andrian@umfiasi.ro (S.A.)

**Keywords:** traditional glass ionomer cement, resin-modified glass ionomer cement, acidic drink, AFM, surface roughness

## Abstract

The aim of the present study was to assess by atomic force microscopy (AFM) the surface roughness of a traditional glass ionomer cement- GIC (Fuji IX GP, GC Corporation, Tokyo, Japan) and a resin modified glass ionomer cement- RMGIC (Vitremer, 3M ESPE, St. Paul, MN, USA) after different immersion regimes on some acidic drinks. Sixteen cylindrical samples having the height of 5 mm and the thickness of 2 mm were obtained from each material and they were divided into two groups: Group I (Fuji IX samples) and Group II (Vitremer samples). Specimens of each group were then randomly divided into 4 subgroups: subgroup A (control)—15 samples were kept in artificial saliva and in the other three subgroups, each having 15 samples the samples were immersed in Coca-Cola (subgroup B), Cappy lemonade and mint (subgroup C) and Fuzetea (subgroup D) for 7 days (subgroups A1–D1), 14 days (subgroups A2–D2), and 21 days (subgroups A3–D3). AFM qualitative and quantitative surface evaluation (mean value of surface roughness parameter, Sa) of each sample was performed. The highest surface roughness was determined when both materials were submerged 14 days in acidic drinks. Traditional GIC was more affected by acidic environment when comparing to RMGIC.

## 1. Introduction

The clinical success of dental restorative biomaterials is mainly based on their properties to resist to mechanical, chemical, and thermal conditions in oral environment. One of the critical factors in ensuring long term survival of the material for direct restoration is their resistance to acidic challenge. There are many sources of acid that can change the pH of oral environment: acid resulting by bacterial biofilm activity, acidic food and beverages, gastric fluid or gastric content, diseases or medication which can affect salivary parameters, occupational exposure [1]. It was demonstrated that acidic condition has a negative influence on tooth structure, but also on materials for restoration as glass ionomer cements, resin-modified glass ionomer cements, polyacid-modified composites (compomers), and composite resins surface characteristics [2,3]. Acidic aggression can lead to a destructive process of restoration surface, with a decrease in wear resistance followed by an increase of surface roughness. Clinical consequences of this process are increased bacterial plaque accumulation, secondary caries lesions formation, gingival irritation, surface discoloration and fatigue failure [4,5]. Bollen et al., reported that the critical surface roughness (Ra) for bacterial colonization is 0.2 μm [6]. Bacterial accumulation, plaque maturation and acidity increase significantly when the surface roughness exceeds 0.2 μm, thus increasing the risk of tooth decay. In addition, surface roughness can directly affect the marginal integrity and wear behavior of the restoration [7]. Several studies have reported that the lowest roughness of glass ionomer cement (GIC) surfaces was found after setting the material relative to the Mylar band [8].

GICs, developed by Wilson and Kent in 1970, contain aluminosilicate strontium glass powder (base) and a soluble polymer (acid) [9,10]. The release of GICs on the market was a consequence of phosphoric acid replacement in the silicate cements with organic chelating acids. Therefore, GIC has been described as a hybrid between silicate and polycarboxylic cements. GIC used for restorations are generally used in occlusal or proximal cavities having low occlusal loads [11,12], in cervical or root cavities, mostly in children and elderly patients, but also in patients having high caries risk. Chemical adhesion to the tooth structure, fluoride release, buffer capacity and tooth protection during dissolution process determined by acidic aggression are some advantages of these materials [13,14].

The aim of the study was to assess using atomic force microscopy the effect of different immersion regimes in various acidic beverages on the surface characteristics of two glass ionomer cements: a traditional GIC and a resin-modified glass ionomer cement (RMGIC). 

## 2. Materials and Methods

The design of the study is presented in Figure 1.

The materials used in this study were Fuji IX GP (GC Corporation, Tokyo, Japan) and Vitremer (3M ESPE, St. Paul, MN, USA). The chemical composition of both materials is presented in Table 1. Fuji IX GP is a conventional type II GIC. It chemically bond to enamel and dentin, which allows the preparation of non-retentive and minimally invasive cavities. Increased flexural and compressive strength ensures this GIC stability, longevity and marginal integrity. Wear resistance is 60% higher compared to other similar materials and it is slightly lower than posterior composites [15]. Vitremer is a RMGIC that stands out as a material which combines the advantages of GIC with those of light-curing systems. The setting reaction relies on three mechanisms: the acid-base reaction initiated when starting the mixing of the powder with the liquid, initiation of free methacrylate radical formation which occurs when the prepared material is subjected to light radiation and the chemical setting which continues in the absence of light curing [16].

### 2.1. Material Specimen Preparation

From each material 60 cylindrical samples having 5 mm in diameter and 2 mm thickness, were obtained by applying the material in plastic conformers. Each mold was placed on glass plates and a celluloid matrix was interposed between the molds and the glass plate in order to obtain a smooth surface and to prevent the air voids formation. After filling the mold, another celluloid matrix was placed on top of the sample and a second glass plate was lightly pressed on top to remove the excess of the material.

The preparation of both materials was carried out accordingly to the manufacturers’ instructions. For Fuji IX the mixture of the powder and the liquid was made on a waxed paper plate using a plastic spatula. The recommended proportions were a measure of powder and a drop of liquid. The powder was divided into two halves. The first half was mixed with the liquid for 10 s, after which the second half was added, continuing to mix for 20 s. The material was taken up 6 min after starting to mix the powder with the liquid. For Vitremer the mixture of the powder with the liquid was also made on a waxed paper plate using a plastic spatula. The recommended proportions were also a measure of powder and a drop of liquid. Subsequently, the samples were light cured for 40 s on both sides through the glass plate to ensure a complete polymerization of the material.

A LED (light emitting diode) light source (Optilight LD MAX, Gnatus, Ribeirão Preto, SP. Brasil) was used for polymerization of resin-modified glass ionomer cement. The source has an ergonomic and compact appearance. Has a well-defined wavelength band (470–480 nm) and a maximum power of 600 mW/cm^2^. This source ensures optimal polymerization up to a depth of 3 mm.

The specimens obtained using Fuji IX GIC were included in group I and the specimens obtained using Vitremer RMGIC were included in group II.

### 2.2. Immersion Protocol in Artificial Saliva and Acidic Drinks

In each group the specimens were randomly divided into 4 subgroups: in subgroup A 15 samples were immersed in artificial saliva (control) and groups B-D consisting of 15 samples each, the samples were submersed on Coca-Cola Zero Green Lemon (SC Coca-Cola HBC Romania SRL, Voluntari, Ilfov, Romania), pH = 2.37, Lemonade by Cappy Lemon and Mint (SC Coca-Cola HBC Romania SRL, Voluntari, Ilfov, Romania), pH = 2.5 and Fuzetea green tea with lime and mint (S.C. Coca-Cola HBC Romania S.R.L., Voluntari, Ilfov, Romania) pH = 3, for 7 days (subgroups A1–D1), for 14 days (subgroups A2–D2), and 21 days (subgroups A3–D3). The pH of these drinks was measured using an electronic pH meter for liquid medium (Testo 206-pH1, Testo Romania, Cluj-Napoca, Romania). The pH was determined at the opening of each beverage bottle from which 15 mL was used, the rest of the content being discarded.

The immersion protocol was the following: in subgroup A all fifteen samples were kept completely immersed in 25 mL of artificial saliva in a tightly closed container at room temperature. The chemical composition of the artificial saliva for one liter of solution and a pH = 6.7, according to the recommendations proposed by Brett et al., is given in Table 2 [17]. The artificial saliva in each container was changed daily. After 7, 14 and 21 days of submersion in saliva the samples were removed, washed with distilled water and dried.

In subgroups B–D, the specimens were completely immersed in acidic beverages for 15 min daily for 7, 14 and 21 days, respectively. The main ingredients of the three acidic beverages are presented in Table 3. The container was stirred continuously during immersion to ensure complete contact of the samples with the acidic medium and to simulate intraoral conditions because while drinking a certain degree of intraoral agitation also takes place. The beverages were renewed after each dip. When they were not immersed in beverages the samples from subgroups B–D of each group were kept in artificial saliva in airtight containers in order to simulate the conditions of intraoral environment.

At the end of the protocol the samples were washed with distilled water and dried with absorbent paper towels.

### 2.3. Surface Characteristics Evaluation

In order to characterize the surfaces of the tested materials the control samples and those subjected to immersion in acidic drinks were analyzed using atomic force microscopy (Nanosurf Easy Scan 2, Nanosurf AG, Liestal, Switzerland), which measures with nanometric resolution the surface characteristics. The microscope has dual lenses and an automatic positioning system for the tip (cantilever) which allows easy scanning of samples of any kind (metallic, non-metallic, organic materials). Technology of the alignment chip in the console allows the simple and fast change of the cantilever without laser adjustments. Thus, the following types of analyzes can be performed: micrography, topography, analysis of thin films or coatings, surface defects and roughness.

The main technical specifications of the microscope scanning heads are described in Figure 2.

For this study the microscope was set to make the measurements in air environment, in static force operating mode. CONTRL cantilever and EZ2 head type were used. Qualitative evaluation of surface condition was performed using three-dimensional images. These images were generated by a signal coming from a photodetector, which is associated with the deflection of the cantilever of the lamp in its interaction with the surface.

For each sample, a quantitative evaluation of surface roughness was also performed on sample areas of 10 × 10 μm using the 10 µm scanner head. In order to determine the average roughness of the analyzed surface, 256 linear scans were performed, obtaining a mean roughness value on each line. The mean arithmetic deviation, Ra, was then calculated based on these values.

### 2.4. Statistical Analyses of the Data

For statistical analyses of the data the IBM SPSS Statistics 28.0.1 program was used. Kolmogorov-Smirnov normality statistical test was used to determine the distribution of data within/between groups. Paired Samples *t*-test, ANOVA and Tukey post hoc statistical tests were used to compare the results between groups and subgroups at *p* < 0.05 significance level.

## 3. Results

Microstructural AFM images of some Fuji IX cement samples in subgroups are presented in Figure 3. The microstructure of the samples in control subgroups reveals the existence of very fine particles on the surface, with a uniform distribution. The 3D images show vertical deviations of up to 1 µm, from which we conclude an initial roughness for all three immersion regimes. The samples in subgroup B are characterized by some micro-channels having evaluated average depth of 0.6 µm. Samples in subgroup C presented a greater number of micro-holes which are wider and can reach up to 2.8 µm depth. The samples in subgroup D revealed also significant changes in surface condition, with variations in profile depth between 0.6–3 µm.

Figure 4 shows AFM microstructural images of Vitremer samples in subgroups. The microstructure of samples in subgroup A reveals the existence of large and irregularly shaped filler particles. 3D images show vertical deviations of the profile, up to 0.2 µm. The samples in subgroup B are characterized by the formation of micro-channels as in the case of the previous material. Moreover, their depth can be assessed as being less than 0.8 µm. The samples in subgroup C revealed a smaller number, finer and less deep micro-depressions of maximum 0.4 µm. The samples in subgroup D presented surface characteristics similar to the samples in subgroup B, but the micro-channels appear more pronounced.

The mean Ra values obtained by quantitative assessment of samples surface using AFM and standard deviation (SD) in groups I and II are presented in Table 4.

Statistical Paired Samples *t*-test was used to compare the results between the groups in each subgroup (significant level < 0.05). Statistically significant results were obtained when comparing the values in group I and group II in all subgroups (Table 6).

In order to compare the results between the subgroups in every group, ANOVA and Tukey post hoc statistical tests were used (significant level < 0.05). The result for multiple comparison in studied groups are represented in Table 7.

In group I there were statistically significant results when comparing subgroups: A1 and B1; A1 and D1; B1 and C1; C1 and D1; A2 and B2; B2 and C2; B2 and D2; A3 and B3. In group II there were statistically significant results when comparing subgroups: A1 and B1; A1 and D1, B1 and C1, B1 and D1; C1 and D1, A2 and B2; A2 and D2; B2 and C2; C2 and D2; A3 and B3.

## 4. Discussion

Foods or soft drinks that contain acidic components, which are often preferred by the young population [18], can cause varying degrees of tooth structure erosion. In oral environment dental restorations are expected to withstand various conditions such as changes in temperature and pH variations that lead to mechanical wear, structural and surface characteristics changes [19]. Therefore, erosion and biodegradation due to the frequent consumption of acidic soft drinks are inevitable also for dental restorations [20]. 

For in vitro studies an environment having low pH will mimic the best in vivo conditions, but the simulated effects are obtained in a long period of time. Therefore, very alkaline or very acidic media can be used to cause rapid microstructural damage [21]. Studies have shown that the erosive potential of an acid solution is determined by its pH, titratable acidity and buffer capacity Most of the soft drinks have a pH of 3.0 or lower [22,23]. It is also the case of the commercial beverages chosen as submersion media in this study: Coca-Cola, Cappy Lemonade and Fuzetea. The pH of Coca-Cola is about 2.37. In addition, this carbonated drink contains a strong inorganic acid, phosphoric acid. The other two studied soft drinks although not carbonated, have also a low pH (Lemonade pH = 2.5 and Fuzetea pH = 3) because they contain organic acids such as ascorbic acid and citric acid. All three acids have the potential to cause the dissolution of GICs [24]. Citric acid is able to chelate ionic components present in glass ionomer-based dental biomaterials and to determine a marked extent of GICs dissolution [25]. Phosphoric acid also can chelate ions to form water-soluble complexes [26]. Exposure to acidic environment could have a detrimental effect on glass ionomer biomaterials. The association between low pH and the presence of these strong acids can lead to an aggressive attack on the surface of the restoration material and therefore increase the surface roughness [27]. Previous studies have reported severe erosion of conventional GICs as result of orange and apple juices action for 3–6 months [28].

Many direct restorations in dental practice are placed on lateral teeth. Physical and mechanical properties of the materials used for direct restoration are essential to ensure the longevity of the restoration [29,30]. In restorative materials wear involves multiple degradation processes, such as chemical corrosion and alteration of mechanical properties with consecutive fatigue of the material. Corrosive wear results from the chemical reaction of the material with the environment and from the mechanical removal of corroded layers. It is associated with the following steps: (i) the process of water absorption, when the restoration is placed in the oral environment, water absorption mainly leading to hydrolytic corrosion of the inorganic filler particles; (ii) the internal diffusion of water through the filling interfaces and micropores, which results in the dissolution of the filling particles and the modification of their reinforcement capacity. If the corroded layer is removed during mastication or brushing, a fresh (new) surface becomes exposed and thus the corrosion cycle continues [31]. Many studies have shown that filler particles tend to be removed from the material and the matrix component decomposes when exposed to low pH environment [32,33]. Roughness can be a consequence of the chemical dissolution of restorative materials caused by exposure to chemical components from acidic beverages and acidic foods. The increase of surface roughness as a result of erosive wear will facilitate bacterial adhesion and secondary caries onset. The roughness of all intraoral surfaces should be approximately of 0.2 µm or less in order to reduce bacterial retention [34]. For both studied materials, an increase in surface roughness can be observed after 7 and 14 days of submersion in acidic beverages probably as a result of the acid attack both on the inorganic fraction and by the dissolution of the organic matrix. Fuji IX presented the surface roughness higher than 0.2 µm after submersion in artificial saliva and acidic beverages, but for Vitremer the mean surface roughness values where lower than 0.2 µm. The surface roughness differences between the two tested materials in this study can be related to their different structure, as it was also reported in other studies [35]. Traditional GIC consist of two main components: a fluoro-aluminosilicate glass powder and an aqueous solution containing polyalkenoic acids which are carboxylic acids. The setting of ionomer glass cements takes place through an acid-base reaction between the polyacrylic acid and the fluoro-aluminosilicate glass particles. The initial setting reaction is a gelation reaction between components, followed by the bonding of unreacted glass particles which act as filler particles in the gelled silicon oxide (SiO_2_) matrix. Traditional GICs have a number of disadvantages, such as brittleness, which subsequently predisposes them to fracture, poor wear resistance and inadequate surface properties which makes them more difficult to finish and polish. In order to overcome the weak mechanical properties of glass ionomers, several changes were introduced in their structure [36]. Key changes include the combination of GIC with self-curing or light-curing resin systems to obtain RMGIC. In addition, the modification of the GIC by incorporating polyvinyl phosphonic acid, reinforcing fibers, bioactive apatite, with or without zirconium, zinc, strontium oxide, silicon particles, aimed to improve the mechanical and physical properties. Filler content influences the chromatic stability, hardness and wear resistance [37]. The behavior of materials in acid challenges depends on the chemical structure. Previous studies have also shown a tendency of surface degradation when these materials were exposed to acidic and alcoholic beverages action [38,39,40]. The consumption of soft drinks over a long period of time can induce erosive wear of restorative materials [41,42,43]. In other studies that have quantified the roughness parameters after immersion in Coca-Cola for longer period of time such 10 days, 20 days and 60 days, respectively it was pointed that the surface roughness of the evaluated glass ionomer cement increased significantly in the first 10 days, decreasing progressively and significantly with the evaluation made after 20 days and 60 days [44]. The impact of a beverage on the materials may be directly related also to the quantity and frequency consumption [45]. In this study in order to evaluate the effects of acidic drinks on glass ionomer-type restorative biomaterials the samples were immersed for 7, 14 and 21 days respectively to simulation the acid attack in the oral environment, following the protocol applied also in other studies [46]. Both traditional glass ionomer cement and resin modified glass ionomer cement presented significant increase of surface roughness after 7 days of submersion in Coca Cola and Fuzetea. After 14 days of submersion in acidic beverages, only Coca Cola led to significantly changings of the traditional glass ionomer cement and resin modified glass ionomer cement surface roughness. Twenty-one days of submersion in Coca Cola affected significantly only the traditional glass ionomer cement surface. Also, other studies demonstrated that resin modified glass ionomer cements are more resistant to dissolution in acidic beverages when comparing to conventional materials [28].

The dissolution process is due to the presence of H^+^ ions in the acidic beverage. The more acidic the drink, the more H^+^ ions will be released and the higher the degree of dissolution of the material will be. Citric acid forms a stable complex with Al^2+^ and Ca^2+^ or Sr^2+^ ions. According to the study by Zaki et al., during immersion in acidic medium, the solution penetrates the cement and the gel matrix increases in size [47]. Hydrogen ions (H^+^) diffuse into the cement and change place with metal cations, which diffuse into the solution based on a decrease in the concentration gradient. The release of metal cations causes an increase in oxygen that is not bound in the glass network near the surface of the material.

Another important factor that influences the surface condition is the internal structure of the material as the size, shape and quantity of the filler particles [48]. Glass particles contain many silanols on the surface and the simultaneous exposure on the cement surface to H^+^ ions will continuously disrupt the Si–O–Si glass bond. The perfect process of dissolving the glass particles will cause the appearance of numerous porous areas on the surface of the cement. This process is very obvious especially for conventional GIC [49]. It can be assumed that softening the organic matrix would favor the dislocation of the filler particles from the matrix by agitation, thus allowing the formation of a surface with increased roughness [50].

This in vitro study aimed to investigate the changes of surface characteristic of two different GICs (conventional and resin-modified) when different regimes of submersion in acidic beverages were simulated. In the study protocol submersion of the tested materials in different liquids can lead to surface deposits formation. Further studies are mandatory to determine the presence and to evaluate such deposits. 

## 5. Conclusions

The qualitative surface evaluation of Fuji IX glass ionomer cement and Vitremer resin-modified glass ionomer cement showed aspects of corrosion after their immersion in acidic beverages (Coca-Cola, Lemonade Cappy and Fuzetea), with the presence of micro-holes unevenly distributed on the surface. The surface effect was much more obviously present on Fuji IX glass ionomer cement and for both materials was much more prominent after 14 days of submersion in acid drinks. Increased surface roughness of both glass ionomer materials was recorded after exposure to acidic challenge represented by Coca-Cola and Fuzetea. The surface of traditional glass ionomer cement was more affected by acidic environment when comparing to resin modified glass ionomer cement.

## Figures and Tables

**Figure 1 biomedicines-10-01755-f001:**
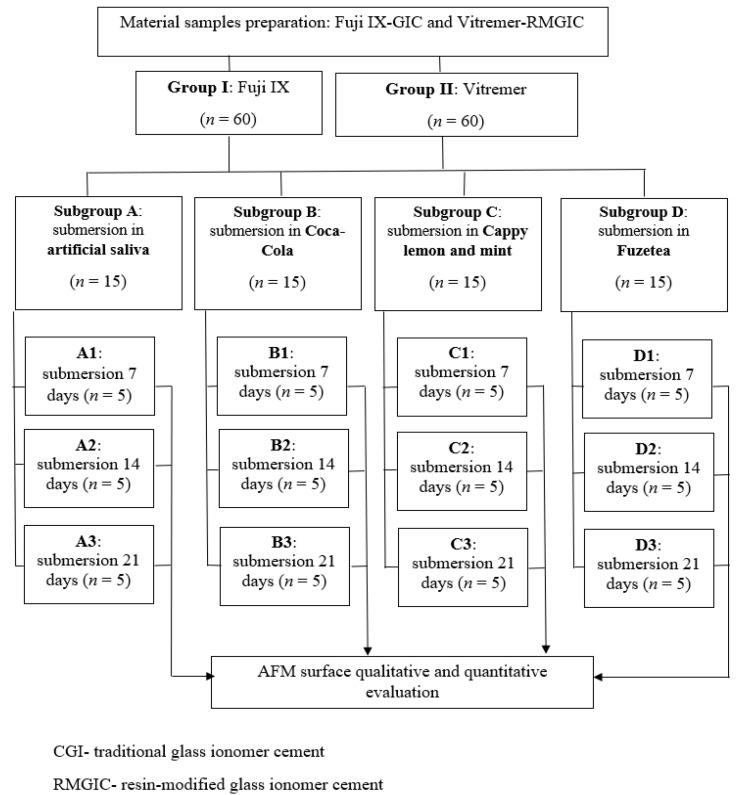
Study design.

**Figure 2 biomedicines-10-01755-f002:**
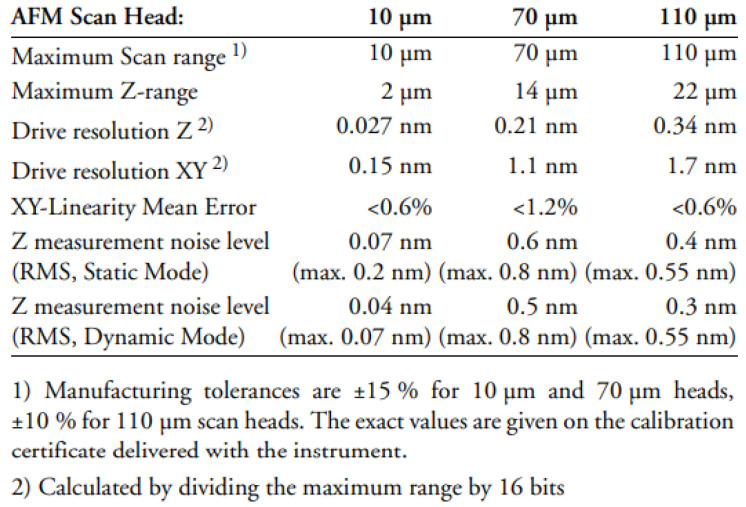
Atomic force microscope technical specifications of scan heads.

**Figure 3 biomedicines-10-01755-f003:**
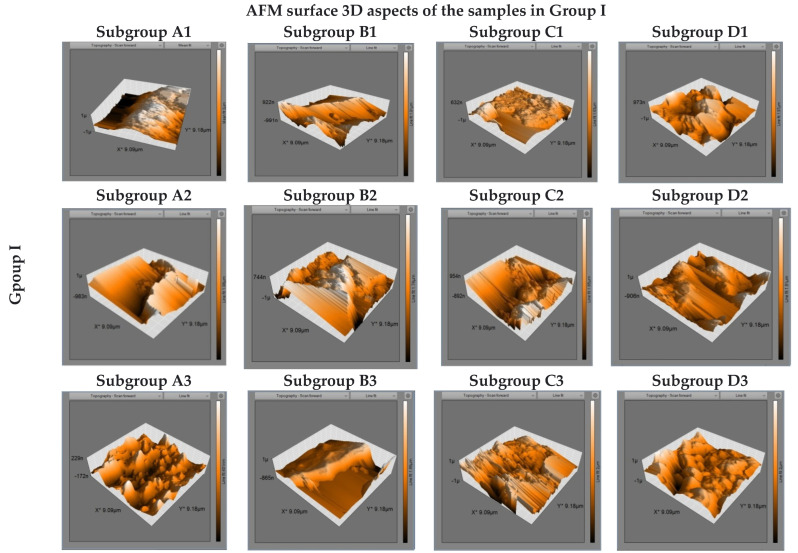
AFM 3D surface topography images of Fuji IX samples after 7, 14, and 21 days of immersion in artificial saliva (subgroups A1–A3), Coca-Cola (subgroups B1–B3), Cappy Lemonade (subgroups C1–C3), and Fuzetea (subgroups D1–D3).

**Figure 4 biomedicines-10-01755-f004:**
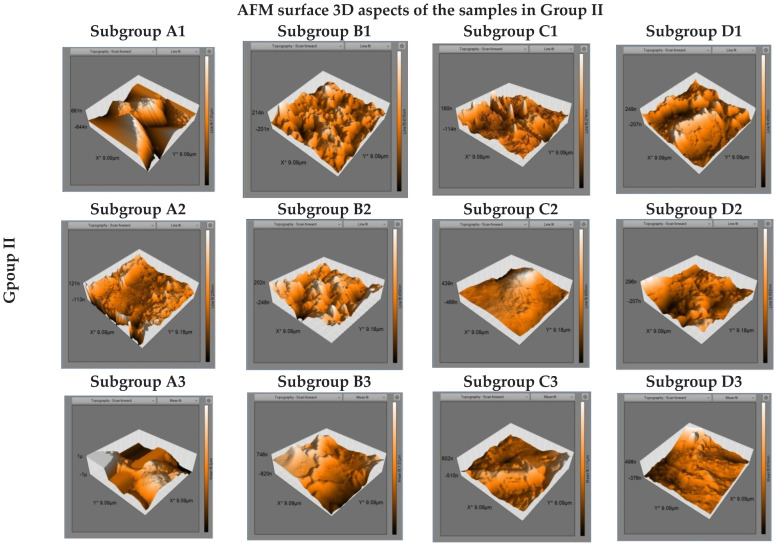
AFM 3D surface topography images of Vitremer samples after 7, 14, and 21 days of immersion in artificial saliva (subgroups A1–A3), Coca-Cola (subgroups B1–B3), Cappy Lemonade (subgroups C1–C3), and Fuzetea (subgroups D1–D3).

**Table 1 biomedicines-10-01755-t001:** Materials used in this study.

Material	Manufacturer	Type	Batch/Shade	Composition
Fuji IX GP	GC Corporation, Tokyo, Japan	Traditionl Glass ionomer Cement	1810021/A3	Powder: Fluro alumino silicate glass, Polyacrylic acid powder.
				Liquid: Polyacrylic acid
				Polybasic carboxylic acid
Vitremer	3M ESPE, St. Paul, MN, USA	Resin-Modified Glass ionomer Cement	N999128/A3	Powder: fluoroaminosilicate glass, potassium persulfate, ascorbic acid.
				Liquid: aqueous solution of a polycarboxylic acid modified with pendant methacrylate groups, water, HEMA, photoinitiators.

Abbreviations: HEMA—Hydroxyethyl methacrylate.

**Table 2 biomedicines-10-01755-t002:** Chemical composition of the artificial saliva solution.

Chemical Compound	Weight (on 1 L)
KCl	1.5 g
NaHCO_3_	1.5 g
NaH_2_PO_4_	0.5 g
KSCN	0.5 g
Lactic acid	0.7 g

**Table 3 biomedicines-10-01755-t003:** Composition of acidic beverages.

Acidic Beverages	Composition
**Coca-Cola Green lemon Zero sugar** (S.C. Coca-Cola HBC România S.R.L., Voluntari,. Ilfov)	Water, Carbon dioxide, Lemon juice from concentrate (0.5%), Phosphoric acid, Sodium citrate, Sugar, Caffeine, Aspartame, Sodium cyclamate, Acesulfame, Sulfate ammoniacal additives (E150d)
**Lemonade by Cappy lemon and mint** (S.C. Coca-Cola HBC România S.R.L., Voluntari, Ilfov)	Water, Lemon juice from concentrate (11%), Sugar, Sodium citrate/E331, Lemon and mint natural aromas, Ascorbic acid, Caroten
**Fuzetea Green tea with Green lemon and mint** (S.C. Coca-Cola HBC România S.R.L., Voluntari, Ilfov)	Water, Sugar, Fructose, Citric acid, Green Tea Extract (0.1%), Lemon juice from concentrate (0.1%), Mint extract (0.01%), Sodium citrate, Ascorbic acid

**Table 4 biomedicines-10-01755-t004:** Mean ra value and standard deviation IN group I and group II.

Subgroups	Mean ± Std. Deviation	
	Group I	Group II
A1	0.34140 ± 0.118423	0.02147 ± 0.012867
B1	0.49267 ± 0.037023	0.04507 ± 0.011919
C1	0.38233 ± 0.061864	0.03100 ± 0.007416
D1	0.40840 ± 0.011975	0.05653 ± 0.006424
A2	0. 47440 ± 0.018995	0.02047 ± 0.008879
B2	0.59200 ± 0.047485	0.05480 ± 0.018876
C2	0.49700 ± 0.056729	0.02887 ± 0.011090
D2	0.47413 ± 0.019291	0.05440 ± 0.005680
A3	0.42533 ± 0.098306	0.02193 ± 0.009169
B3	0.48313 ± 0.022077	0.03500 ± 0.007010
C3	0.47440 ± 0.018995	0.02707 ± 0.011622
D3	0.39513 ± 0.028553	0.03007 ± 0.007667

Kolmogorov-Smirnov normality test showed that in all groups the data were normal distributed (*p* > 0.05) (Table 5).

**Table 5 biomedicines-10-01755-t005:** Kolmogorov-Smirnov normality test result.

		Group I—07 Days	Group I—14 Days	Group I—21 Days	Group II—7 Days	Group II—14 Days	Group II—21 Days
N		60	60	60	60	60	60
Normal Parameters ^ab^	Mean	0.42270	0.50938	0.02852	0.03852	0.03963	0.02852
	Std. Deviation	0.093089	0.062241	0.010023	0.016650	0.019413	0.010023
Most Extreme Differences	Absolute	0.170	0.250	0.077	0.074	0.085	0.077
	Positive	0.106	0.250	0.077	0.062	0.085	0.077
	Negative	–0.170	–0.099	–0.058	–0.074	–0.071	–0.058
Kolmogorov-Smirnov Z		1.314	1.933	0.600	0.576	0.657	0.600
Asymp. Sig, (2-tailed)		0.063	0.001	0.864	0.895	0.781	0.864

^a^ Test distribution is Normal; ^b^ Calculated from data.

**Table 6 biomedicines-10-01755-t006:** Paired samples *t*-test result of comparing groups I and II in subgroups.

	Paired Diffrences					t	df	Sig(2-Tailed)
	Mean	Std. Deviation	Std. Error Mean	95% Confidence Interval of the Diffrences				
				Lower	Upper			
Pair 1 IA1–IIA1	0.319933	0.116402	0.030055	0.255472	0.384395	10.645	14	0
Pair 2 IB1–IIB1	0.4476	0.040771	0.010527	0.425022	0.470178	42.519	14	0
Pair 3 IC1–IIC1	0.351333	0.062256	0.016074	0.316857	0.38581	21.857	14	0
Pair 4 ID1–IID1	0.351867	0.014667	0.003787	0.343744	0.359989	92.914	14	0
Pair 5 IA2–IIA2	0.453933	0.023998	0.006196	0.440643	0.467223	73.258	14	0
Pair 6 IB2–IIB2	0.5372	0.059514	0.015366	0.504242	0.570158	34.959	14	0
Pair 7 IC2–IIC2	0.468133	0.054797	0.014148	0.437788	0.498479	33.087	14	0
Pair 8 ID2–IID2	0.419733	0.020232	0.005224	0.408529	0.430938	80.347	14	0
Pair 9 IA3–IIA3	0.4034	0.093493	0.02414	0.351625	0.455175	16.711	14	0
Pair 10 IB3–IIB3	0.448133	0.023213	0.005994	0.435278	0.460988	74.769	14	0
Pair 11 IC3–IIC3	0.447333	0.02562	0.006615	0.433145	0.461521	67.624	14	0
Pair 12 ID3–IID3	0.365067	0.029651	0.007656	0.348646	0.381487	47.684	14	0

**Table 7 biomedicines-10-01755-t007:** Tukey post-hoc statistical test result of comparing the subgroups in groups I and II.

Subgroups	Groups		Subgroups	Groups		Subgroups	Groups	
	I	II		I	II		I	II
A1-B1	0.000 *	0.000 *	A2–B2	0.000 *	0.000 *	A3–B3	0.001 *	0.001 *
A1-C1	0.386	0.056	A2–C2	0.403	0.242	A3–C3	0.413	0.413
A1-D1	0.000 *	0.000 *	A2–D2	1.000	0.000 *	A3–D3	0.077	0.077
B1-C1	0.000 *	0.002 *	B2–C2	0.000 *	0.000	B3–C3	0.088	0.088
B1-D1	0.891	0.000 *	B2–D2	0.000 *	1.000	B3–D3	0.448	0.448
C1-D1	0.004 *	0.003 *	C2–D2	0.393	0.000 *	C3–D3	0.800	0.800

* The mean difference is significant at the 0.05 level.

## Data Availability

Not applicable.

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
