# Peer review of "Conventional and Resin-Modified Glass Ionomer Cement Surface Characteristics after Acidic Challenges"

_biomedicines, 2022, doi:10.3390/biomedicines10071755_

Round 1
Reviewer 1 Report
The work is very interesting, but due to the study of surfaces only, it is difficult to talk about what is happening in the materials apart from a neat discussion of the various possibilities.
Ultimately, we learn no more than roughness changes, but we don't know why, and are you sure you are not seeing surface deposits?
Without water absorption vs / other liquids and weight loss, in my opinion the work is incomplete, but I believe that the authors have the results for subsequent publications.
In the title:
cement resistance to acidic challenges
It is not "resistance", it has no such physical quantity, unless a collection of different, but since you are only investigating surface roughness, why a misleading title?
the time of polymerization (light) ?
Line 138-157
the general description of the method is unnecessary, but the parameter for the performed test is missing
Line 133
the samples were washed with distilled water and dried 133 with absorbent paper towels.
considering the liquid's aging ingredients, I am not sure if this cleans the surface. Can you confirm that we can test the material without deposits?
Author Response
Thank you for the time and effort to review the manuscript.
Please find bellow point-to-point answers to each comment of yours.
|
Comment |
Action |
|
The work is very interesting, but due to the study of surfaces only, it is difficult to talk about what is happening in the materials apart from a neat discussion of the various possibilities. |
Due to the complexity of chemical interaction of GICs with liquid (acidic) environment (dissolution and precipitation processes), this study aimed just to evaluate the effect of different regimes of submersion in acidic beverages on surface characteristics like surface roughness. Further studies will evaluate the presence of the surface deposits in the conditions of oral cavity that we have simulated in this in vitro study. We have added these aspects in the main text (lines 318-323). |
|
In the title: It is not "resistance", it has no such physical quantity, unless a collection of different, but since you are only investigating surface roughness, why a misleading title? |
We have changed the title to reflect better the aim, method and the results (we changed „resistance to acidic challenges‟ with „surface characteristics after acidic challenges‟) (line 1-3) |
|
the time of polymerization (light) ? |
The time of polymerization process to complete the setting of Vitremer resin-modified glass ionomer cement was 40 seconds (line97) |
|
Line 138-157 |
AFM 10µm scanner head was used in this study. The mean arithmetic deviation, Ra parameter was reported for each sample as a result of 256 linear scan (lines 158-157). We have added the parameters of AFM device used in this study (lines 149-150) |
|
Line 133 |
Our further studies will evaluate the presence of the surface deposits in clinical conditions that we simulated in this in vitro study. We have added this aspect in the main text (lines 318-323). |
Reviewer 2 Report
This paper reports a useful study of effect of acidic beverages on commercial glass-ionomer cements (conventional and resin-modified. As such, it is a confirmatory study, but has neglected two key references, which describe the first detailed study of these beverages on glass-ionomers. The two references which should be consulted and included are:
M. Aliping-McKenzie, et al, The physical properties of conventional and resin-modified glass-ionomer dental cements stored in saliva, proprietary acidic beverages, saline and water, Biomaterials, 24, 4063-4069, (2003).
M. Aliping-McKenzie, at al, The long-term effect of Coca-Cola and fruit juices on the surface hardness of glass-ionomers and compomers, J. Oral Rehabil. 31, 1046-1052, (2004).
In addition, the title of the current paper should be changed (Replace the word "Traditional" with "Conventional"). Line 51 ignores the fact that GICs (including Fuji IX GP used in the study) are strontium-based and do not contain calcium. This omission also appears in line 230.
In lines 297 and 298, there are typographical errors for Al3+, Ca2+ and Sr2+. Please correct this.
Once these changes are made, the paper will be suitable for publication.
Author Response
Thank you for the time and effort to review the manuscript.
Please find bellow point-to-point answers to each comment of yours.
|
Comment |
Action |
|
This paper reports a useful study of effect of acidic beverages on commercial glass-ionomer cements (conventional and resin-modified. As such, it is a confirmatory study, but has neglected two key references, which describe the first detailed study of these beverages on glass-ionomers. The two references which should be consulted and included are: M. Aliping-McKenzie, et al, The physical properties of conventional and resin-modified glass-ionomer dental cements stored in saliva, proprietary acidic beverages, saline and water, Biomaterials, 24, 4063-4069, (2003). M. Aliping-McKenzie, at al, The long-term effect of Coca-Cola and fruit juices on the surface hardness of glass-ionomers and compomers, J. Oral Rehabil. 31, 1046-1052, (2004).
|
We have added the references in the manuscript- references no. 28, 47 (lines 401, 441). |
|
In addition, the title of the current paper should be changed (Replace the word "Traditional" with "Conventional"). |
We have changed in the title "traditional" with "conventional") (lines 1-3). |
|
Line 51 ignores the fact that GICs (including Fuji IX GP used in the study) are strontium-based and do not contain calcium. This omission also appears in line 230. |
We have made the correction at line 51 by removing calcium of being a component of GICs powder (line 51). We have removed from the main text the comment regarding chelation reaction of citric acid with calcium ions from GICs (lines 230-231). |
|
In lines 297 and 298, there are typographical errors for Al3+, Ca2+ and Sr2+. Please correct this. |
We have made the corrections (lines 302-303) |